# TOWARD GENERALIZABLE DEBLURRING: LEVERAGING MASSIVE BLUR PRIORS WITH LINEAR ATTENTION FOR REAL-WORLD SCENARIOS

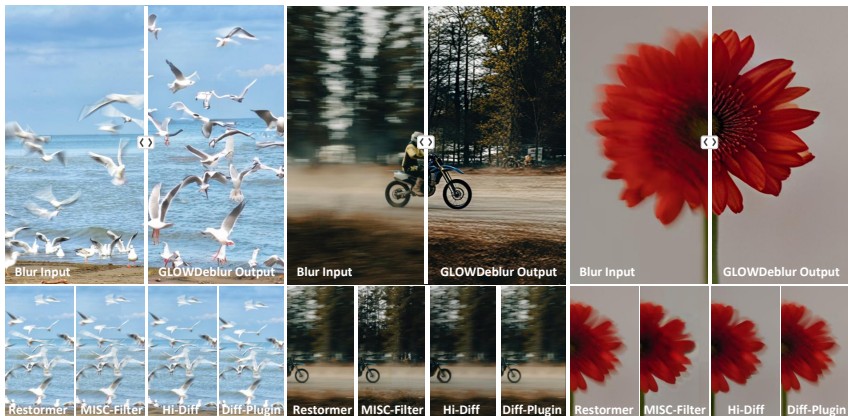

(a) Deblurring Results on Challenging Real-World Scenes

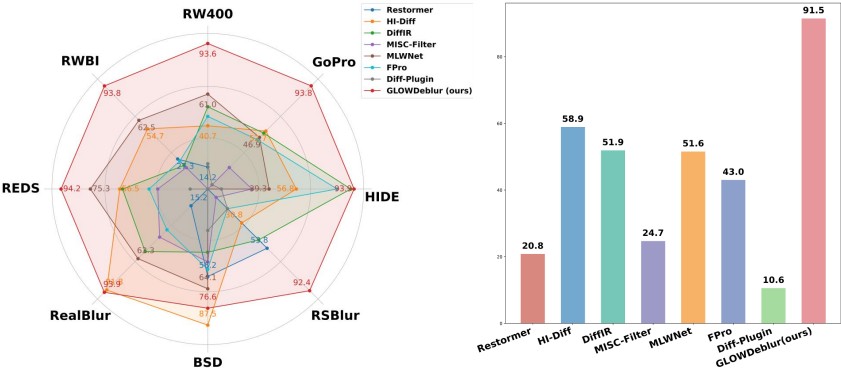

(b) Performance Comparison across Multiple Datasets

Figure 1: (a) **Visual comparison on challenging real-world images:** our GLOWDeblur effectively restores a wide range of blur patterns, while prior methods often fail in complex scenarios. (b) **Quantitative comparison on diverse benchmarks:** the left plot shows dataset scores computed by ranking methods on each metric and averaging across metrics; the right plot reports average model scores across all datasets, highlighting the strong generalization ability of GLOWDeblur.

## ABSTRACT

Image deblurring has advanced rapidly with deep learning, yet most methods exhibit poor generalization beyond their training datasets, with performance dropping significantly in real-world scenarios. Our analysis shows this limitation stems from two factors: datasets face an inherent trade-off between realism and coverage of diverse blur patterns, and algorithmic designs remain restrictive, as pixel-wise losses drive models toward local detail recovery while overlooking structural and semantic consistency, whereas diffusion-based approaches, though perceptually strong, still fail to generalize when trained on narrow datasets with simplistic strategies. Through systematic investigation, we identify blur pattern diversity as the decisive factor for robust generalization and propose Blur Pattern Pretraining (BBP), which acquires blur priors from simulation datasets and transfers them

through joint fine-tuning on real data. We further introduce Motion and Semantic Guidance (MoSeG) to strengthen blur priors under severe degradation, and integrate it into **GLOWDeblur**, a **G**eneralizable rea**L**-w**O**rld light**W**eight **Deblur** model that combines convolution-based pre-reconstruction & domain alignment module with a lightweight diffusion backbone. Extensive experiments on six widely-used benchmarks and two real-world datasets validate our approach, confirming the importance of blur priors for robust generalization and demonstrating that the lightweight design of GLOWDeblur ensures practicality in real-world applications.

# 1 INTRODUCTION

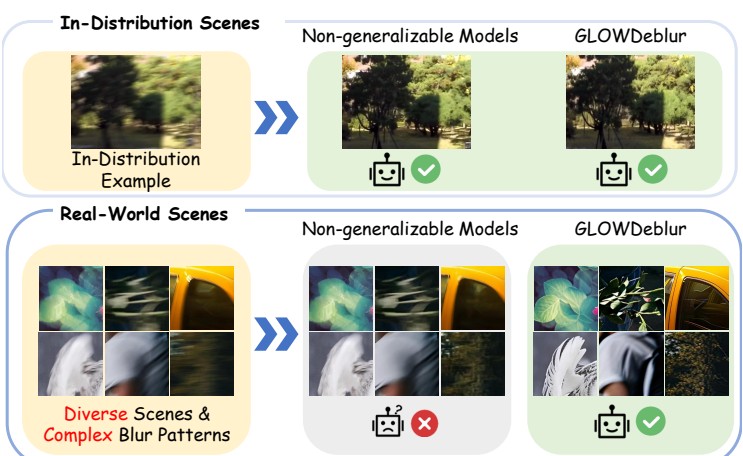

Figure 2: Challenges for Real-World Generalization

In recent years, image deblurring has made significant progress with the rapid development of deep learning. A variety of high-quality datasets Nah et al. (2017); Shen et al. (2019); Nah et al. (2019); Rim et al. (2020); Zhong et al. (2020); Rim et al. (2022); Lee et al. (2024) and advanced algorithms Chen et al. (2023); Liu et al. (2024a); Gao et al. (2024) have been proposed, achieving impressive performance across benchmarks. However, these advances have not resolved a central limitation: most approaches are trained and evaluated on a limited set of datasets, leading to overfitting to their domain characteristics and specific blur patterns. As a result, their generalization performance drops noticeably when applied to real-world scenarios, where blur is inherently more diverse and complex. As illustrated in Fig. 1, where three real-world cases show that current state-of-the-art methods fail to deliver satisfactory restorations not only in complex scenes but also in a visually simple case, reflecting the inherent challenges of real-world blur. Moreover, substantial gaps exist among current datasets, and naive mixed-dataset training not only fails to improve generalization but often degrades performance on the original benchmarks. This raises a central challenge: how to effectively organize existing datasets and design deblurring frameworks that can substantially improve generalization, enabling models to robustly handle the diverse and complex blur patterns encountered in real-world conditions.

Through systematic investigation, we find that this limitation arises from two key aspects: dataset construction and algorithmic design. Current datasets face inherent constraints, making it difficult to achieve both realism and comprehensive coverage of blur patterns. Synthetic datasets such as GoPro Nah et al. (2017) and REDS Nah et al. (2019) allow large-scale training but diverge from real-world distributions, while real-captured datasets like RealBlur Rim et al. (2020) and RS-Blur Rim et al. (2022) improve realism but remain limited in blur diversity and scene coverage. Even simulation-based datasets such as GSBlur Lee et al. (2024) still differ significantly from real-world degradations. Consequently, substantial gaps remain both across datasets and between synthetic datasets and real-world blur, hindering models trained on a single dataset from achieving robust generalization. Beyond the data, algorithmic choices also impose important constraints. Models trained with pixel-wise losses (e.g., MSE) favor local details but overlook global structure and semantics, leading to smooth outputs with poor generalization Zamir et al. (2022); Liu et al. (2024a); Gao et al. (2024); Zhou et al. (2024). Diffusion models leverage strong priors for perceptually better

results, but training on narrow datasets with simple strategies limits their ability to capture diverse blur patterns Chen et al. (2023); Xia et al. (2023); Liu et al. (2024b).

Based on these observations, we first conduct a systematic analysis of dataset biases in deblurring. While prior research has largely emphasized the realism of blur Rim et al. (2020); Zhong et al. (2020); Rim et al. (2022), we find that the diversity and coverage of blur patterns—such as their orientation and spatial distribution—are critical factors behind the gaps observed both across datasets and between datasets and real-world blur. Motivated by this finding, we propose **BBP** (**B**lur **P**attern **P**retraining): a data-centric strategy where models are first pretrained on large-scale simulation datasets with comprehensive blur patterns to acquire strong blur priors, and are then jointly fine-tuned on real-captured datasets. This process enables the model to leverage blur priors to bridge dataset gaps, ultimately improving both robustness and applicability in real-world deblurring.

In terms of algorithm design, diffusion models offer strong prior modeling and the ability to integrate heterogeneous data sources, making them well suited for generalizable deblurring. However, their high complexity and resource demands hinder deployment in real-world applications that require real-time efficiency, such as autonomous driving and mobile photography. To address this, we propose **GLOWDeblur**, a **G**eneralizable rea**L**-w**O**rld light**W**eight **Deblur** model that combines a convolution-based pre-reconstruction & domain-alignment module with a lightweight diffusion model, which employs a Deep Compression AutoEncoder and Linear Attention. To further strengthen the model's ability to handle diverse and complex real-world blur, we incorporate motion guidance and cross-modal semantic captions as complementary signals, enabling the model to better adapt to varied blur patterns and recover severely degraded regions by leveraging the generative capacity of diffusion models. GLOWDeblur is trained with our Blur Pattern Pretraining (BBP) strategy and extensively evaluated on six widely used benchmarks and two real-world datasets. Results show that GLOWDeblur achieves superior cross-dataset and real-world generalization, underscoring blur priors as the key to real-world deblurring.

In summary, this work makes the following contributions:

• **Revealing the role of blur patterns.** We systematically analyze dataset biases and reveal that the diversity and coverage of blur patterns, rather than realism alone, are the decisive factors behind cross-dataset gaps. Learning blur priors and leveraging them as guidance is shown to be essential for achieving robust and quantifiable generalization.

• **Data- and model-level priors for generalization.** We introduce Blur Pattern Pretraining (BBP), a data-centric strategy that first learns blur priors from large-scale simulation datasets and then jointly fine-tunes on real-captured datasets. In parallel, we propose Motion and Semantic Guidance (MoSeG) to reinforce blur priors and alleviate structural and semantic degradation under severe blur.

• **A generalizable real-world deblurring model.** We propose GLOWDeblur, a diffusion-based framework that balances efficiency and effectiveness, achieving strong performance across six benchmarks and two real-world datasets. Beyond results, it also serves as a practical testbed to validate our insights and demonstrate real-world applicability.

## 2 MOTIVATION

### 2.1 LIMITATIONS OF EXISTING MODELS IN REAL-WORLD BLUR SCENARIOS

Although recent methods have achieved remarkable progress, they still exhibit fundamental limitations, particularly in generalizing to diverse real-world blur patterns. As illustrated in Fig. 1, across three representative real-world scenes, current state-of-the-art methods fail to deliver satisfactory restorations beyond the training distribution, not only under complex scenes but even in visually simple ones. Fig. 2 further reinforces this observation: although existing methods handle in-distribution blur reasonably well, they suffer severe failures when confronted with the diverse and complex scenes and blur patterns of real-world scenarios. This indicates that current approaches rely heavily on dataset-specific distributions rather than learning transferable representations of blur.

These observations motivate us to examine the roots of the generalization gap, revealing that explicitly modeling blur-pattern priors and organizing training data to capture their diversity are crucial for robust real-world deblurring. Guided by these insights, we design improved training strategies and a lightweight model that generalize effectively across diverse scenes and blur patterns (Fig. 2 GLOWDeblur), thereby overcoming the limitations of existing methods.

## 2.2 DATASET BIAS AND BLUR PATTERN DISCREPANCIES

To understand the generalization gap, we conducted a series of cross-dataset experiments using Restormer as a representative backbone. Models were first trained individually on six widely used datasets and one simulation-based dataset constructed via 3D Gaussian Splatting, and then evaluated across all datasets. As shown in Tab. 1, models trained on one dataset degrade notably on others, underscoring a substantial cross-dataset distribution gap.

Table 1: Cross-dataset results (PSNR/SSIM) reveal severe generalization gaps, with red indicating the best in-dataset and blue the second-best cross-dataset result. Avg column reports mean PSNR/SSIM across datasets.

| Training Set \ Test Set | GoPro | HIDE | REDS | RealBlur | BSD | RSBlur | GSBlur | Avg |
|---|---|---|---|---|---|---|---|---|
| GoPro (Synthetic) | 32.92 / 0.94 | 31.22 (↓0.40)/ 0.92 | 26.93 (↓7.46) / 0.82 | 28.96 (↓3.13) / 0.88 | 24.43 (↓9.32) / 0.90 | 29.30 (↓3.68) / 0.86 | 24.94 (↓6.43) / 0.82 | 28.39 / 0.88 |
| HIDE (Synthetic) | 32.60 (↓0.32) / 0.94 | 31.62 / 0.93 | 26.68 (↓7.71) / 0.83 | 27.76 (↓4.33) / 0.86 | 25.34 (↓8.41) / 0.85 | 27.77 (↓5.21) / 0.83 | 23.40 (↓7.97) / 0.80 | 27.88 / 0.86 |
| REDS (Synthetic) | 26.21 (↓6.71) / 0.83 | 24.42 (↓7.20) / 0.80 | 34.39 / 0.94 | 28.72 (↓3.37) / 0.86 | 28.90 (↓4.85) / 0.84 | 28.09 (↓4.89) / 0.87 | 24.42 (↓6.95) / 0.80 | 27.88 / 0.85 |
| RealBlur (Real) | 24.50 (↓8.42) / 0.82 | 23.60 (↓8.02) / 0.81 | 25.85 (↓8.54) / 0.79 | 32.09 / 0.92 | 28.78 (↓4.97) / 0.91 | 29.65 (↓3.33) / 0.87 | 24.92 (↓6.45) / 0.81 | 27.06 / 0.85 |
| BSD (Real) | 27.27 (↓5.65) / 0.86 | 26.27 (↓5.35) / 0.85 | 28.20 (↓6.19) / 0.84 | 29.64 (↓2.45) / 0.89 | 33.75 / 0.96 | 30.45 (↓2.53) / 0.89 | 26.78 (↓4.59) / 0.85 | 28.91 / 0.88 |
| RSBlur (Real) | 27.55 (↓5.37) / 0.87 | 25.79 (↓5.83) / 0.84 | 28.08 (↓6.31)/0.84 | 30.41 (↓1.68) / 0.89 | 30.85 (↓2.90) / 0.94 | 32.98 / 0.93 | 27.63 (↓3.74) / 0.86 | 29.04 / 0.88 |
| GSBlur (Simulated) | 28.51 (↓4.41) / 0.90 | 26.12 (↓5.50) / 0.87 | 30.29 (↓4.10) / 0.90 | 30.06 (↓2.03) / 0.91 | 31.24 (↓2.51) / 0.94 | 32.01 (↓0.97) / 0.92 | 31.37 / 0.92 | 29.94 / 0.91 |

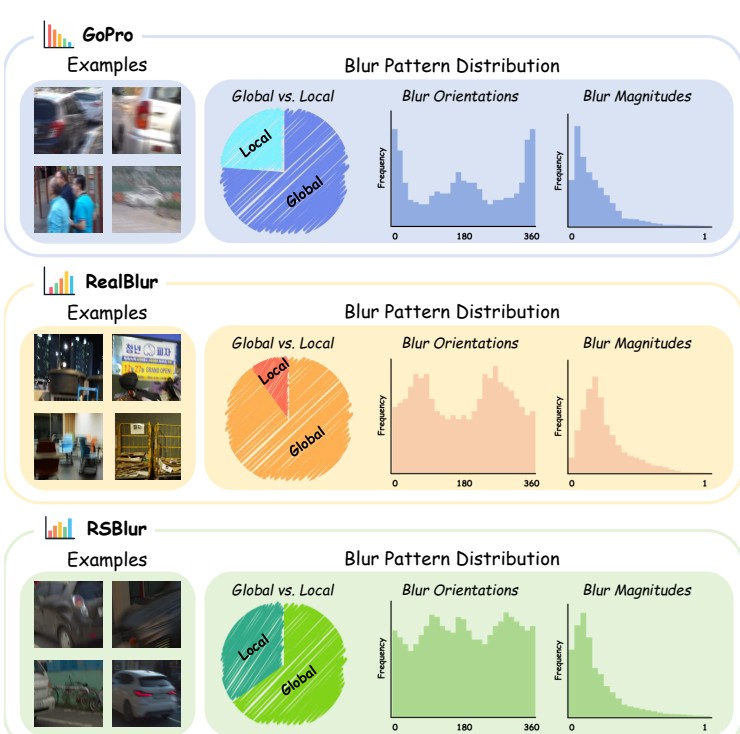

Figure 3: Illustration of dataset-specific blur patterns, highlighting notable distribution differences.

Importantly, Our cross-dataset experiments further reveal two important observations. First, these gaps exist not only between synthetic and real datasets, but also within the same category (synthetic vs. synthetic or real vs. real), indicating that beyond realism there exist deeper sources of mismatch. Second, despite limited realism in both scenes and blur in GSBlur, its broad coverage of blur patterns allows models trained on it to achieve relatively stronger cross-dataset robustness. Collectively, these results highlight that blur pattern diversity, insufficiently recognized in prior work, plays a dominant role in causing the significant cross-dataset gap.

To validate this insight, we conducted a fine-grained analysis of blur characteristics across datasets. As shown in Fig. 3, their blur patterns differ markedly in orientation, magnitude, and locality. In particular, GoPro is dominated by horizontal blur while RealBlur is primarily vertical, yet prior work has often attributed their discrepancy only to differences in realism.

In summary, our analysis shows that dataset bias in deblurring arises primarily from blur pattern mismatches, thereby motivating our exploration of both data-centric strategies to mitigate cross-dataset gaps and algorithmic frameworks that exploit blur priors for robust real-world generalization.

## 3 METHODOLOGY

### 3.1 BLUR PATTERN PRETRAINING (BPP)

Since blur pattern diversity is key to generalization, we propose Blur Pattern Pretraining (BPP) that uses datasets with broad blur coverage to enable models to learn blur pattern priors, thereby mitigating distribution gaps and enhancing both performance and generalization. Table 2 illustrates this effectiveness using Restormer as a testbed. Pretraining on GSBlur, which offers diverse blur patterns despite limited realism, and then fine-tuning on RealBlur, BSD, and RSBlur (a,b,c) consistently boosts both in-dataset performance and cross-dataset generalization over direct training. And Naïve mixed training (d) not only fails to achieve strong results across the three datasets but also degrades performance due to the pronounced gaps between them, whereas applying BBP before mixing (e) effectively mitigates these gaps and yields comprehensive improvements on all datasets.

Table 2: Performance comparison on RealBlur-J, BSD, and RSBlur under different training settings.

| No. | Training set | BBP | RealBlur-J | BSD | RSBlur |
|---|---|---|---|---|---|
| (a) | RealBlur | ✓ | 32.26 (↑0.17) / 0.93 (↑0.01) | 29.76 (↓3.99) / 0.92 (↓0.04) | 30.28 (↓2.70) / 0.89 (↓0.04) |
| (b) | BSD | ✓ | 29.95 (↓2.14) / 0.90 (↓0.02) | 34.21 (↑0.46) / 0.96 (±0.00) | 31.15 (↓1.83) / 0.90 (↓0.03) |
| (c) | RSBlur | ✓ | 30.63 (↓1.46) / 0.90 (↓0.02) | 31.22 (↓2.53) / 0.95 (↓0.01) | 33.69 (↑0.71) / 0.94 (↑0.01) |
| (d) | RealBlur + BSD + RSBlur | ✗ | 30.83 (↑1.26) / 0.89 (↓0.03) | 31.99 (↓1.76) / 0.95 (↓0.01) | 31.24 (↓1.74) / 0.90 (↓0.03) |
| (e) | RealBlur + BSD + RSBlur | ✓ | 32.11 (↑0.02) / 0.93 (↑0.01) | 33.62 (↓0.13) / 0.96 (±0.00) | 33.65 (↑0.67) / 0.94 (↑0.01) |
| | **Best same-dataset performance** | – | **32.09 / 0.92** | **33.75 / 0.96** | **32.98 / 0.93** |

Because BBP proves highly effective in bridging distribution gaps and improving performance, we incorporate it into the training of GLOWDeblur. As illustrated in Fig. 4, the model first performs BBP on a simulated dataset that provides comprehensive blur pattern coverage, enabling GLOWDeblur to internalize essential blur-related knowledge and priors. In the subsequent stage, the model is fine-tuned on multiple real-captured datasets, adapting the learned knowledge to close cross-dataset gaps and further enhance both generalization and restoration performance.

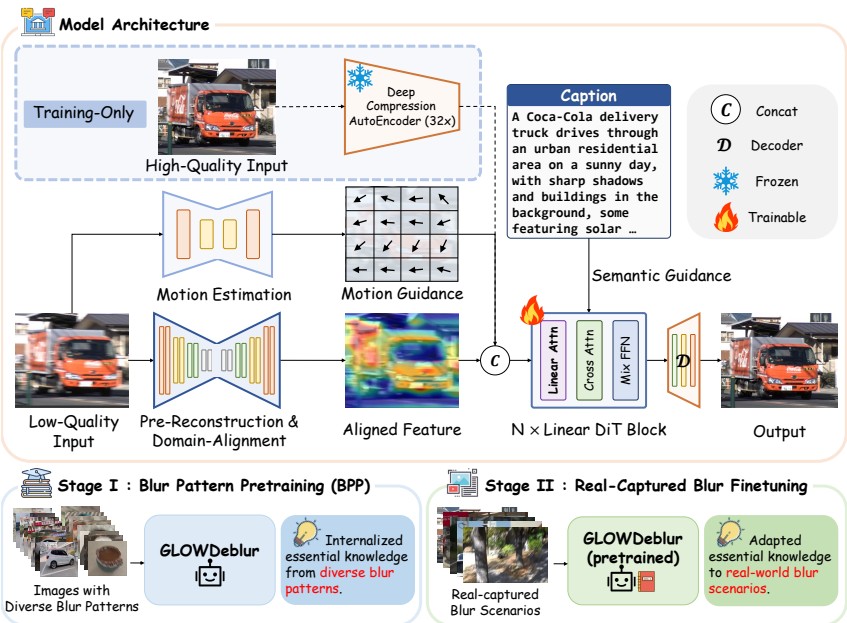

Figure 4: Overview of GLOWDeblur. The framework integrates a Pre-Reconstruction & Domain-Alignment module with a lightweight diffusion framework, guided by motion maps and cross-modal text semantics. Training involves pre-training on datasets with diverse blur patterns, followed by joint fine-tuning on real-captured datasets.

### 3.2 MOTION AND SEMANTIC GUIDANCE (MOSEG)

While BBP equips models with transferable blur priors, challenges remain under severe or highly diverse blur, where structural cues are ambiguous and low-level details are heavily lost. To address this, we introduce Motion and Semantic Guidance (MoSeG), a conditional design that explicitly reinforces blur priors during inference and training.

**Motion Guidance (MoG):** To strengthen the guidance of blur priors, we integrate a motion estimation module. Estimation of motion trajectories provides a direct way to characterize blur pat-

terns and enhance the model's ability to generalize across diverse degradations. The blur can be modeled as the accumulation of displaced sharp pixels along estimated trajectories: $B(p_0) = \frac{1}{N}\sum_{n=0}^{N-1} L_s(p_0 + \Delta P_{t_n})$, where $L_s$ is the latent sharp image and $\Delta P_{t_n}$ the motion offset at $t_n$.

Following prior work on motion offset estimation Zhang et al. (2021), we adopt a lightweight encoder–decoder that extracts hierarchical features and predicts dense motion fields $\Delta P$. These offsets are concatenated with blurred-image features and fed into the deblurring network as motion cues.

**Semantic Guidance (SeG):** In severely blurred regions where structural details are lost, we inject high-level semantics as conditional signals to unleash the cross-modal capacity of diffusion models. Specifically, using QwenVL-2.5-7B Bai et al. (2025), we generate detailed captions describing objects, scenes, context, and other high-level attributes, and feed their embeddings into Linear DiT blocks, enabling the recovery of heavily degraded regions.

### 3.3 LIGHTWEIGHT PRE-ALIGNED LINEAR DIFFUSION FRAMEWORK

Real-world deblurring applications, ranging from autonomous driving to mobile photography, demand models that are both highly efficient and compact. To this end, we design a lightweight framework that integrates a Pre-Reconstruction & Domain-Alignment module with a Deep Compression AutoEncoder and Linear DiT blocks, achieving both efficiency and strong performance.

**Pre-Reconstruction & Domain-Alignment module:** a conventional UNet architecture that provides coarse restoration and aligned intermediate representations, reducing the burden on the diffusion backbone. To keep the design lightweight, we follow the philosophy of Chen et al. (2022), simplifying architectures with two key modifications. First, nonlinear activations such as GELU are replaced with a SimpleGate, where feature maps are split and fused via element-wise product:

$$\text{SimpleGate}(X, Y) = X \odot Y, \tag{1}$$

preserving gating capacity at negligible cost. Second, channel attention is reformulated as Simplified Channel Attention (SCA), which aggregates global context through pooled descriptors and reweights channels without redundant nonlinearities:

$$SCA(X) = X \odot W_{pool}(X). \tag{2}$$

Together, these modifications substantially reduce computation while retaining representational power.

**Lightweight Diffusion with Deep Compression AutoEncoder and Linear Attention:** Latent diffusion operates by compressing images into a latent space via an AutoEncoder and applying a DiT for diffusion within this space, where the computational cost is largely influenced by the compression ratio and the complexity of attention mechanisms. To meet real-world efficiency demands, we adopt lightweight adaptations inspired by Xie et al. (2024); Xiaohui Li (2025).

While mainstream designs typically adopt an $8\times$ AutoEncoder for latent compression, we employ a more aggressive $32\times$ Deep Compression AutoEncoder. With sufficient performance maintained, this design reduces the number of tokens and significantly lowers memory and computation.

Traditional Diffusion Transformers (DiTs) adopt the standard softmax attention mechanism with quadratic complexity $O(N^2)$, which is computationally expensive. we further replace the quadratic self-attention in the DiT with a linear variant, reducing the complexity to $O(N)$.

Given query $\mathbf{Q} \in \mathbb{R}^{N \times d}$, key $\mathbf{K} \in \mathbb{R}^{N \times d}$, and value $\mathbf{V} \in \mathbb{R}^{N \times d}$, the linear attention output is defined as:

$$O_i = \sum_{j=1}^{N} \frac{\text{ReLU}(Q_i)\,\text{ReLU}(K_j)^\top V_j}{\sum_{j=1}^{N} \text{ReLU}(Q_i)\,\text{ReLU}(K_j)^\top} = \frac{\text{ReLU}(Q_i)\Big(\sum_{j=1}^{N} \text{ReLU}(K_j)^\top V_j\Big)}{\text{ReLU}(Q_i)\Big(\sum_{j=1}^{N} \text{ReLU}(K_j)^\top\Big)}. \tag{3}$$

Instead of computing attention weights for every query–key pair, the shared terms $\sum_{j=1}^{N} \text{ReLU}(K_j)^\top V_j \in \mathbb{R}^{d \times d}$ and $\sum_{j=1}^{N} \text{ReLU}(K_j)^\top \in \mathbb{R}^{d \times 1}$ are computed only once, resulting in a lightweight and effective Linear DiT Block.

Finally, we adopt a lightweight fusion strategy that concatenates shallow convolutional features with motion guidance and latent representations before processing by the Linear DiT block, effectively integrating complementary cues while preserving efficiency.

## 4 EXPERIMENTS

### 4.1 EXPERIMENT SETTINGS

Training proceeds in two stages. First, BBP is performed on simulated datasets, including GSBlur (3D Gaussian Splatting with randomized camera trajectories) and an augmented subset of LSDIR Li et al. (2023) (by simply adding Gaussian and motion blur at different levels). Although less realistic, these datasets provide broad blur pattern coverage . Second, we jointly fine-tune on GoPro, HIDE, REDS, RealBlur, BSD, and RSBlur to align these priors with real-world distributions and enhance restoration quality. More implementation details are given in the Appendix B.

Table 3: Quantitative comparison with state-of-the-art deblurring methods on six widely used benchmarks. Higher values indicate better performance for ↑ metrics, and lower values for ↓.

| Dataset | Metric | Restormer* Zamir et al. (2022) | HI-Diff Chen et al. (2023) | DiffIR Xia et al. (2023) | MISC-Filter Liu et al. (2024a) | MLWNet Gao et al. (2024) | FPro Zhou et al. (2024) | Diff-Plugin Liu et al. (2024b) | Ours |
|---|---|---|---|---|---|---|---|---|---|
| GoPro Nah et al. (2017) | PSNR ↑ | 33.07 | 33.33 | 33.20 | 34.10 | 24.60 | 33.05 | 25.64 | 25.21 |
| | SSIM ↑ | 0.943 | 0.964 | 0.963 | 0.969 | 0.83 | 0.943 | 0.793 | 0.787 |
| | MANIQA ↑ | 0.353 | 0.492 | 0.535 | 0.458 | 0.497 | 0.518 | 0.346 | 0.538 |
| | LIQE ↑ | 1.455 | 1.350 | 1.589 | 1.172 | 1.353 | 1.491 | 1.092 | 1.502 |
| | NRQM ↑ | 4.748 | 5.047 | 5.051 | 4.339 | 4.750 | 4.915 | 3.886 | 5.252 |
| | CLIP-IQA ↑ | 0.243 | 0.250 | 0.258 | 0.214 | 0.257 | 0.250 | 0.190 | 0.239 |
| | PI ↓ | 5.308 | 5.159 | 5.135 | 5.464 | 5.363 | 5.151 | 6.202 | 4.846 |
| | BRISQUE ↓ | 46.715 | 46.418 | 46.721 | 46.095 | 49.638 | 49.121 | 51.018 | 39.732 |
| | NIQE ↓ | 5.534 | 5.504 | 5.466 | 5.461 | 5.650 | 5.377 | 6.146 | 5.120 |
| | ILNIQE ↓ | 33.354 | 32.710 | 33.408 | 33.071 | 32.535 | 32.701 | 42.474 | 26.464 |
| HIDE Shen et al. (2019) | PSNR ↑ | 31.81 | 31.46 | 31.55 | 31.66 | 23.95 | 30.70 | 23.95 | 24.12 |
| | SSIM ↑ | 0.933 | 0.945 | 0.947 | 0.946 | 0.819 | 0.921 | 0.763 | 0.763 |
| | MANIQA ↑ | 0.453 | 0.535 | 0.592 | 0.498 | 0.509 | 0.572 | 0.362 | 0.583 |
| | LIQE ↑ | 1.113 | 1.621 | 1.977 | 1.236 | 1.392 | 1.803 | 1.061 | 1.788 |
| | NRQM ↑ | 3.916 | 5.613 | 6.104 | 4.731 | 5.129 | 6.028 | 4.323 | 6.210 |
| | CLIP-IQA ↑ | 0.187 | 0.227 | 0.229 | 0.179 | 0.224 | 0.215 | 0.158 | 0.212 |
| | PI ↓ | 6.302 | 4.837 | 4.354 | 5.278 | 5.085 | 4.308 | 5.773 | 4.244 |
| | BRISQUE ↓ | 52.830 | 43.605 | 41.045 | 45.919 | 48.277 | 42.970 | 40.716 | 36.687 |
| | NIQE ↓ | 6.558 | 5.254 | 4.803 | 5.318 | 5.348 | 4.624 | 5.528 | 4.673 |
| | ILNIQE ↓ | 36.248 | 31.246 | 29.788 | 29.985 | 30.702 | 28.224 | 40.744 | 24.176 |
| REDS Nah et al. (2019) | PSNR ↑ | 34.207 | 25.760 | 26.78 | 27.58 | 27.60 | 26.96 | 26.27 | 26.21 |
| | SSIM ↑ | 0.938 | 0.779 | 0.819 | 0.832 | 0.851 | 0.840 | 0.771 | 0.770 |
| | MANIQA ↑ | 0.536 | 0.630 | 0.626 | 0.607 | 0.647 | 0.613 | 0.518 | 0.642 |
| | LIQE ↑ | 1.435 | 2.530 | 2.293 | 2.065 | 2.664 | 2.097 | 1.520 | 2.570 |
| | NRQM ↑ | 4.886 | 6.849 | 7.014 | 6.541 | 6.954 | 6.809 | 6.384 | 7.352 |
| | CLIP-IQA ↑ | 0.271 | 0.305 | 0.287 | 0.268 | 0.322 | 0.269 | 0.228 | 0.351 |
| | PI ↓ | 5.335 | 3.379 | 3.391 | 3.546 | 3.293 | 3.481 | 4.160 | 3.035 |
| | BRISQUE ↓ | 43.364 | 28.061 | 26.452 | 30.115 | 31.448 | 28.533 | 27.867 | 25.695 |
| | NIQE ↓ | 5.660 | 3.899 | 3.973 | 3.894 | 3.851 | 3.966 | 4.620 | 3.694 |
| | ILNIQE ↓ | 29.305 | 23.784 | 23.269 | 23.083 | 22.369 | 23.236 | 27.088 | 19.357 |
| RealBlur-J Rim et al. (2020) | PSNR ↑ | 31.131 | 29.15 | 25.37 | 33.88 | 33.84 | 27.90 | 26.25 | 27.55 |
| | SSIM ↑ | 0.917 | 0.890 | 0.825 | 0.938 | 0.941 | 0.873 | 0.79 | 0.811 |
| | MANIQA ↑ | 0.472 | 0.629 | 0.571 | 0.602 | 0.615 | 0.544 | 0.467 | 0.613 |
| | LIQE ↑ | 2.356 | 2.646 | 1.949 | 2.386 | 2.578 | 1.735 | 1.243 | 2.439 |
| | NRQM ↑ | 5.150 | 5.870 | 5.517 | 5.365 | 5.685 | 5.361 | 4.283 | 5.633 |
| | CLIP-IQA ↑ | 0.262 | 0.279 | 0.247 | 0.251 | 0.274 | 0.212 | 0.208 | 0.274 |
| | PI ↓ | 5.235 | 4.651 | 4.934 | 5.011 | 4.869 | 5.013 | 5.965 | 4.787 |
| | BRISQUE ↓ | 49.636 | 46.799 | 40.207 | 46.610 | 48.970 | 42.742 | 42.401 | 35.895 |
| | NIQE ↓ | 5.708 | 5.182 | 5.258 | 5.341 | 5.370 | 5.256 | 5.963 | 5.128 |
| | ILNIQE ↓ | 34.999 | 33.380 | 31.848 | 34.550 | 33.588 | 32.580 | 37.317 | 27.548 |
| BSD Zhong et al. (2020) | PSNR ↑ | 30.410 | 28.66 | 27.97 | 29.53 | 28.82 | 26.64 | 27.67 | 29.56 |
| | SSIM ↑ | 0.923 | 0.907 | 0.885 | 0.923 | 0.910 | 0.885 | 0.862 | 0.893 |
| | MANIQA ↑ | 0.571 | 0.565 | 0.362 | 0.510 | 0.536 | 0.461 | 0.427 | 0.568 |
| | LIQE ↑ | 2.325 | 2.452 | 1.048 | 1.742 | 2.132 | 1.479 | 1.296 | 2.348 |
| | NRQM ↑ | 4.927 | 5.806 | 4.634 | 4.772 | 5.229 | 5.141 | 4.433 | 5.096 |
| | CLIP-IQA ↑ | 0.279 | 0.283 | 0.188 | 0.234 | 0.264 | 0.179 | 0.195 | 0.282 |
| | PI ↓ | 5.800 | 5.189 | 6.100 | 5.819 | 5.560 | 5.934 | 6.422 | 5.589 |
| | BRISQUE ↓ | 47.363 | 39.518 | 24.113 | 46.358 | 46.388 | 29.108 | 36.347 | 40.514 |
| | NIQE ↓ | 5.799 | 5.464 | 6.469 | 5.758 | 5.721 | 5.881 | 6.569 | 5.455 |
| | ILNIQE ↓ | 42.040 | 38.708 | 31.513 | 41.328 | 40.506 | 37.435 | 50.182 | 39.383 |
| RSBlur Rim et al. (2022) | PSNR ↑ | 29.27 | 29.47 | 22.48 | 29.98 | 30.91 | 26.19 | 27.82 | 28.85 |
| | SSIM ↑ | 0.864 | 0.875 | 0.651 | 0.887 | 0.818 | 0.833 | 0.821 | 0.820 |
| | MANIQA ↑ | 0.442 | 0.452 | 0.362 | 0.420 | 0.415 | 0.398 | 0.441 | 0.533 |
| | LIQE ↑ | 1.342 | 1.124 | 1.048 | 1.069 | 1.111 | 1.018 | 1.015 | 1.404 |
| | NRQM ↑ | 3.769 | 3.817 | 4.634 | 3.523 | 3.642 | 5.520 | 4.357 | 5.597 |
| | CLIP-IQA ↑ | 0.262 | 0.246 | 0.188 | 0.204 | 0.248 | 0.170 | 0.169 | 0.236 |
| | PI ↓ | 6.427 | 6.851 | 6.100 | 6.820 | 7.065 | 6.296 | 6.677 | 4.980 |
| | BRISQUE ↓ | 52.768 | 50.286 | 24.113 | 54.119 | 58.433 | 39.250 | 21.942 | 30.677 |
| | NIQE ↓ | 6.522 | 7.348 | 6.469 | 6.943 | 7.532 | 7.533 | 6.840 | 5.292 |
| | ILNIQE ↓ | 32.349 | 37.794 | 31.513 | 36.354 | 39.651 | 36.035 | 41.705 | 25.833 |

We evaluate methods on six real-captured datasets for cross-dataset generalization, and on RWBI Zhang et al. (2020) and our collected RWBlur400 for real-world applicability, as they cover complex real-world scenes and diverse real-world blur patterns.

Overall, We employ both reference-based and no-reference metrics to comprehensively assess deblurring performance. For reference-based evaluation, PSNR and SSIM are used. To better capture perceptual quality, we further adopt a diverse set of no-reference quality metrics, including

MANIQA Yang et al. (2022), LIQEZhang et al. (2023), NRQM Ma et al. (2017), CLIP-IQA Wang et al. (2023), PI, BRISQUE, NIQE, and ILNIQE, providing a thorough assessment of quality.

## 4.2 COMPARISONS WITH STATE OF THE ARTS

We compare GLOWDeblur with state-of-the-art approaches from two categories. The first includes recent deblurring-specific methods such as HI-Diff, MISCFilter, and MLWNet. The second covers general restoration frameworks such as Restormer (and Restormer* retrained under our pipeline), DiffIR, FPro, and Diff-Plugin. Both categories contain a mix of diffusion-based and non-diffusion baselines, allowing a fair and comprehensive evaluation.

### 4.2.1 CROSS-DATASET GENERALIZATION

We evaluate cross-dataset generalization on six widely used datasets. As shown in Fig. 3, GLOWDeblur mitigates cross-dataset distribution gaps, achieving strong deblurring performance and high-quality restoration with competitive fidelity. Restormer* also achieves good fidelity across different datasets. Fig. 5 provides qualitative comparisons (with more results in the Appendix E), showing that GLOWDeblur robustly handles complex blur patterns and delivers high-quality restorations.

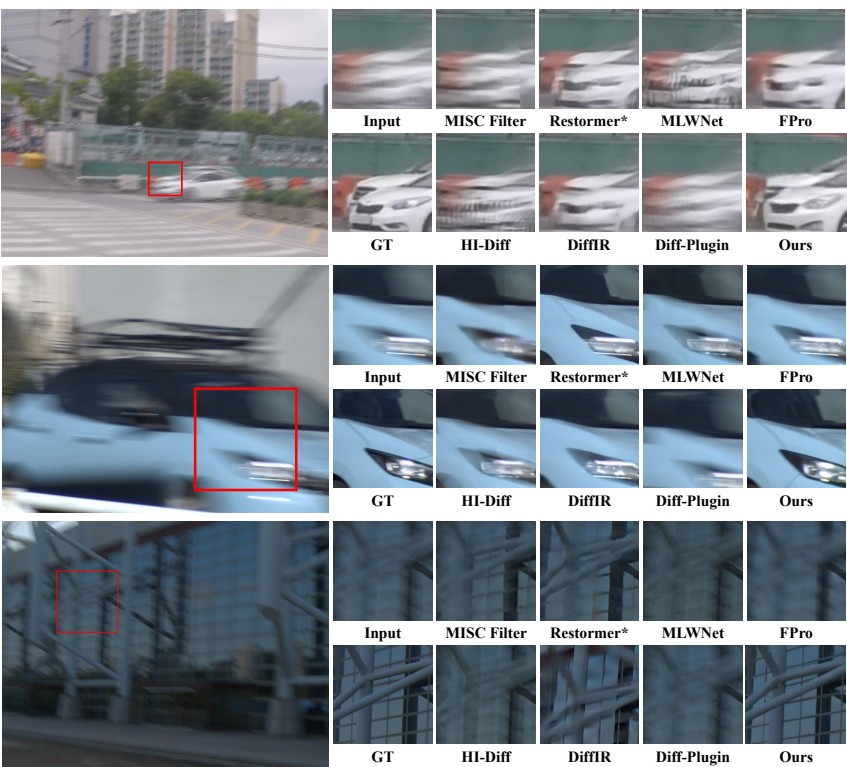

Figure 5: Qualitative comparison on GoPro, BSD, and RSBlur (From top to bottom). GLOWDeblur effectively handles diverse blur patterns with high-quality restorations.

Table 4: Quantitative comparison with SOTA deblur models across real-world datasets RWBI and RWBlur400. Higher values are better for ↑ metrics, lower for ↓. Since these datasets lack ground-truth annotations, only no-reference metrics are reported.

| Dataset | Metric | Restormer Zamir et al. (2022) | Restormer* Zamir et al. (2022) | HI-Diff Chen et al. (2023) | DiffIR Xia et al. (2023) | MISC-Filter Liu et al. (2024a) | MLWNet Gao et al. (2024) | FPro Zhou et al. (2024) | Diff-Plugin Liu et al. (2024b) | Ours |
|---|---|---|---|---|---|---|---|---|---|---|
| RWBI Zhang et al. (2020) | MANIQA ↑ | 0.501 | 0.522 | 0.554 | 0.494 | 0.492 | 0.565 | 0.483 | 0.460 | 0.635 |
| | LIQE ↑ | 1.846 | 2.180 | 2.875 | 1.807 | 2.150 | 3.068 | 1.771 | 1.371 | 3.732 |
| | NRQM ↑ | 5.433 | 5.608 | 5.879 | 5.457 | 5.079 | 6.185 | 5.474 | 5.028 | 6.393 |
| | CLIP-IQA ↑ | 0.257 | 0.283 | 0.372 | 0.256 | 0.301 | 0.424 | 0.236 | 0.234 | 0.474 |
| | PI ↓ | 5.291 | 5.133 | 4.993 | 5.353 | 5.468 | 4.459 | 5.182 | 5.670 | 4.789 |
| | BRISQUE ↓ | 39.945 | 39.192 | 40.674 | 42.319 | 43.271 | 37.403 | 39.581 | 40.488 | 36.625 |
| | NIQE ↓ | 5.682 | 5.545 | 5.589 | 5.831 | 5.793 | 4.886 | 5.550 | 5.988 | 4.796 |
| | ILNIQE ↓ | 40.760 | 37.569 | 37.047 | 41.304 | 38.705 | 34.259 | 40.208 | 47.048 | 33.271 |
| RWBlur400 | MANIQA ↑ | 0.509 | 0.530 | 0.490 | 0.493 | 0.455 | 0.517 | 0.481 | 0.497 | 0.604 |
| | LIQE ↑ | 1.746 | 1.893 | 2.041 | 1.983 | 1.780 | 2.136 | 1.844 | 1.808 | 2.390 |
| | NRQM ↑ | 4.956 | 5.471 | 5.411 | 5.747 | 4.901 | 5.736 | 5.783 | 5.660 | 6.746 |
| | CLIP-IQA ↑ | 0.333 | 0.352 | 0.367 | 0.339 | 0.305 | 0.362 | 0.313 | 0.360 | 0.459 |
| | PI ↓ | 4.788 | 4.734 | 4.990 | 4.578 | 5.236 | 4.461 | 4.520 | 4.649 | 3.786 |
| | BRISQUE ↓ | 41.083 | 40.620 | 43.704 | 34.578 | 46.365 | 41.043 | 31.025 | 30.259 | 27.956 |
| | NIQE ↓ | 4.971 | 4.817 | 5.229 | 4.751 | 5.204 | 4.575 | 4.609 | 4.777 | 4.576 |
| | ILNIQE ↓ | 31.732 | 31.796 | 34.968 | 32.232 | 34.699 | 32.491 | 32.749 | 33.989 | 29.809 |

### 4.2.2 REAL-WORLD EVALUATION

We evaluate real-world performance on RWBI and collected RWBlur400 datasets. As shown in Table 4, GLOWDeblur consistently outperforms state-of-the-art baselines, demonstrating superior performance under real-world degradations. Restormer* also achieves significant improvements compared to its original version. Fig. 6 presents qualitative comparisons, where existing methods fail under severe blur, but GLOWDeblur produces clear and reliable restorations in real-world scenarios. (more results in the Appendix E)

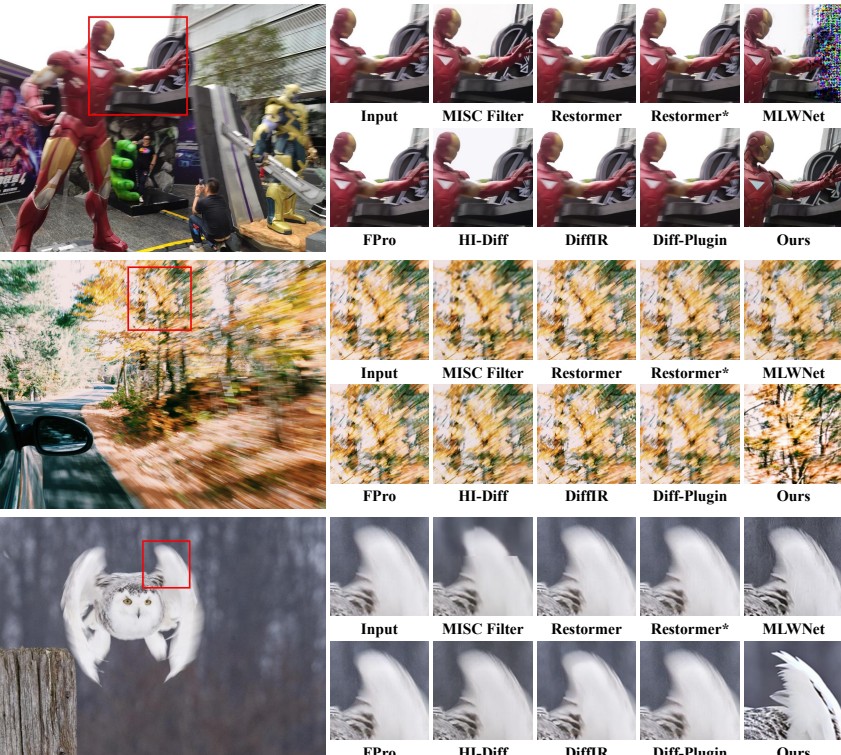

Figure 6: Comparison with SOTA deblur models on real-world datasets RWBI and RWBlur400.

### 4.3 ABLATION STUDIES

We conduct ablation studies on REDS and RSBlur (Tab. 5). The Pre-Reconstruction & Domain-Alignment module provides consistent gains (a,b) by stabilizing representations and easing the diffusion backbone. Motion guidance enhances blur priors with trajectory cues (b,c), while semantic guidance introduces high-level semantics to recover severely degraded regions (c,d). Replacing BBP with naïve mixed-data training causes clear drops (d,e), confirming its role in bridging cross-dataset gaps. Overall, these results validate the effectiveness of both BBP and the model components.

Table 5: Ablation studies on REDS and RSBlur. Gray indicates the settings of GLOWDeblur.

| Dataset | No. | $N_{\text{Pre-aline}}$ | $G_{\text{motion}}$ | $G_{\text{text}}$ | BBP | Mix all data | MANIQA ↑ | LIQE ↑ | NRQM ↑ | BRISQUE ↓ |
|---------|-----|------|------|------|------|------|------|------|------|------|
| REDS | (a) | × | × | × | ✓ | × | 0.565 | 1.817 | 5.968 | 34.297 |
| | (b) | ✓ | × | × | ✓ | × | 0.605 | 2.099 | 6.250 | 32.230 |
| | (c) | ✓ | ✓ | × | ✓ | × | 0.635 | 2.480 | 7.340 | 27.810 |
| | (d) | ✓ | ✓ | ✓ | ✓ | × | **0.642** | **2.570** | **7.352** | **25.695** |
| | (e) | ✓ | ✓ | ✓ | × | ✓ | 0.608 | 2.255 | 6.014 | 31.794 |
| RSBlur | (a) | × | × | × | ✓ | × | 0.463 | 1.132 | 4.392 | 43.854 |
| | (b) | ✓ | × | × | ✓ | × | 0.484 | 1.181 | 4.715 | 39.940 |
| | (c) | ✓ | ✓ | × | ✓ | × | 0.526 | 1.369 | 5.421 | 32.578 |
| | (d) | ✓ | ✓ | ✓ | ✓ | × | **0.533** | **1.404** | **5.597** | **30.677** |
| | (e) | ✓ | ✓ | ✓ | × | ✓ | 0.495 | 1.283 | 5.057 | 41.510 |

## 5 CONCLUSION

In this work, we identify blur pattern diversity as key to generalization and propose GLOWDeblur, a lightweight diffusion-based framework that integrates Blur Pattern Pretraining (BBP) and Motion & Semantic Guidance (MoSeG), achieving state-of-the-art performance on multiple synthetic and real-world datasets with substantially stronger generalization than existing models.

**Limitations.** To ensure lightweight design, our modules are simplified, sacrificing some performance. A finer trade-off between efficiency and accuracy may further improve generalization with modest parameter increases.

ETHICS STATEMENT

Our work adheres to ethical standards throughout model design, training, and dataset collection. All datasets are sourced from publicly available and legally compliant repositories, and no personally identifiable or sensitive information is included. Data annotation was conducted responsibly with clear guidelines to ensure fairness and reduce bias. The model design and training strictly follow principles of transparency and reproducibility, without any practices that may cause harm or infringe on privacy. Overall, this study complies with ethical norms and aims to contribute positively to the research community.

REPRODUCIBILITY STATEMENT

We provide all necessary details to ensure the reproducibility of our work. Specifically, the experimental designs, dataset usage strategies, and implementation in Appendix B. In addition, the training schedules, hyperparameters, and evaluation protocols are included to allow faithful replication of our results. The codebase and scripts are also released in the supplementary materials to further facilitate reproducibility.

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

# A  RELATED WORKS

## A.1  IMAGE DEBLURRING

Image deblurring has long been a fundamental problem in low-level vision. Earlier methods mainly relied on handcrafted priors and optimization-based formulations, such as gradient sparsity, edge sharpness, or statistical constraints Krishnan et al. (2011); Pan et al. (2016). While these approaches provided valuable insights, their strong reliance on manually designed assumptions made them inadequate for handling complex and diverse real-world blur. With the rise of deep learning, researchers have shifted toward data-driven architectures, enabling significant improvements in both restoration quality and efficiency. In this work, we focus on the latest generation of learning-based approaches that have recently achieved state-of-the-art performance in deblurring and general image restoration.

Task-specific deblurring architectures Tsai et al. (2022); Liu et al. (2024a); Gao et al. (2024); Chen et al. (2023) have been extensively explored. Non-diffusion approachesintroduce specialized designs tailored for motion blur removal. Tsai et al. (2022) employs directional strip-based attention to capture region-specific blur orientations and magnitudes efficiently. Liu et al. (2024a) leverages motion-adaptive collaborative filtering to handle spatially variant motion in real-world settings. Gao et al. (2024) integrates multi-scale prediction with learnable wavelet transforms to preserve frequency and directional continuity. On the diffusion side, Chen et al. (2023) designs a compact latent diffusion model with hierarchical integration to generate blur-aware priors for regression-based restoration. These specialized models typically achieve strong performance in blur removal but often generalize poorly when facing unseen blur patterns.

Beyond specialized models, general-purpose restoration frameworks Wang et al. (2022); Zamir et al. (2022); Xia et al. (2023); Liu et al. (2024b); Zhou et al. (2024) have also been widely applied to deblurring. Non-diffusion methods demonstrate strong versatility across tasks. Wang et al. (2022) employs locally enhanced window attention to scale to high-resolution restoration, Zamir et al. (2022) introduces channel-wise self-attention for efficient global context modeling , and Zhou et al. (2024) incorporates frequency prompting to guide restoration across different degradations. Diffusion-based methods Xia et al. (2023); Liu et al. (2024b) further extend general restoration: Xia et al. (2023) integrates compact priors into efficient denoising diffusion, and Liu et al. (2024b) introduces lightweight task-specific plugin modules to adapt pre-trained diffusion models across diverse low-level vision tasks. Compared with task-specific designs, these general frameworks exhibit stronger cross-task robustness and generalization, though they often lag behind specialized models in task-optimized fidelity.

Despite these advances, most existing approaches still struggle with generalization in real-world scenarios. While task-specific methods achieve strong performance under their training distributions, they often fail to transfer across diverse blur patterns. General-purpose frameworks, though more robust across degradations, tend to sacrifice task-optimized fidelity. Overall, systematic investigation into real-world generalization for deblurring remains limited, leaving a critical gap that our work aims to address.

## A.2  DEBLURRING DATASETS

Progress in image deblurring has been closely tied to the availability of datasets. Yet, constructing suitable datasets is inherently challenging, as it involves balancing realism, diversity, and scalability. Synthetic datasets Nah et al. (2017); Shen et al. (2019); Nah et al. (2019) such as GoPro Nah et al. (2017), HIDE Shen et al. (2019), and REDS Nah et al. (2019) have been the dominant benchmarks for years. They are generated through pipelines that average or interpolate high-frame-rate videos to simulate camera exposure, offering large-scale paired data at relatively low cost. Such datasets have enabled rapid progress by providing standardized benchmarks, but the blur they simulate often deviates from real imaging processes. As a result, models trained on synthetic data may perform well in-distribution but fail to capture the irregular, spatially variant blur patterns observed in the wild.

To reduce this gap, real-captured datasets Rim et al. (2020; 2022); Zhong et al. (2020) have been developed. RealBlur Rim et al. (2020), BSD Zhong et al. (2020), and RSBlur Rim et al. (2022) adopt specialized imaging systems—such as beam-splitter setups or synchronized multi-camera rigs—to capture geometrically aligned pairs of blurred and sharp images. These datasets provide authen-

tic motion and defocus blur, more faithfully reflecting the complexity of real-world degradations. However, the hardware cost and collection complexity are significant, limiting the dataset scale and diversity. Even with substantial effort, it remains nearly impossible to comprehensively cover the range of blur magnitudes, orientations, and scene dynamics encountered in real scenarios.

Recently, simulation-based datasets such as GSBlur Lee et al. (2024) have been proposed to improve diversity and controllability. By reconstructing 3D scenes with Gaussian Splatting and rendering them under randomized camera trajectories, GSBlur generates blur patterns beyond traditional frame-averaging pipelines. While this controllability broadens the degradation space, simulated blur still lacks the photometric and structural fidelity of real imaging, leaving a clear gap to real-captured datasets.

In summary, synthetic datasets are abundant but unrealistic, real-captured ones are authentic but costly and narrow, and simulation-based ones offer diversity but lack realism. No dataset achieves both scale and fidelity, creating distribution gaps that cause models to overfit specific blur patterns and degrade sharply under unseen conditions. This underscores the need for strategies that explicitly address blur diversity and distribution mismatch, motivating our work.

### A.3  DIFFUSION MODELS

Diffusion Models (DMs) Esser et al. (2024); Ho et al. (2020); Rombach et al. (2022) have recently emerged as powerful generative priors, synthesizing data from Gaussian noise through iterative denoising. Their success in image generation has inspired a series of applications in deblurring. In the context of deblurring, DiffIR Xia et al. (2023) and HI-Diff Chen et al. (2023) adopt diffusion-based priors with a two-stage training strategy to better capture blur statistics, More recently, IDBlau Wu et al. (2024)leverages implicit diffusion to augment blur patterns under controllable settings, effectively enriching training data for downstream deblurring models.

Despite their effectiveness, most of these approaches remain computationally expensive. Large-scale pretrained diffusion models  Yu et al. (2024); Esser et al. (2024); Ai et al. (2024)possess billions of parameters, which, while offering strong generative priors, impose prohibitive training and inference costs that limit deployment in real-world scenarios like autonomous driving and mobile imaging. This challenge has motivated efforts to develop lightweight alternatives. For example, Xie et al. (2024) proposes a linear-attention-based diffusion transformer that achieves high efficiency without sacrificing quality, demonstrating that architectural re-design and aggressive compression can bring diffusion models closer to practical deployment. Similarly,  Xiaohui Li (2025) and related works explore simplified diffusion formulations tailored for image restoration. Nonetheless, the exploration of lightweight diffusion for deblurring remains limited, leaving an open question of how to balance generalization, restoration fidelity, and efficiency under real-world constraints.

## B  IMPLEMENTATION DETAILS

Our model is trained in two stages. First, Blur Pattern Pretraining (BBP) is performed for 10k iterations on a synthetic mixture of GSBlur and an augmented subset of LSDIR, where Gaussian and motion blur of varying levels are added to enrich pattern diversity. The model is then fine-tuned until convergence on a combined real-captured dataset including GoPro, HIDE, REDS, RealBlur, BSD, and RSBlur, aligning the learned priors with real-world distributions. Training is conducted using the Adam optimizer with an initial learning rate of 1e-4 and a batch size of 12×8. All experiments are implemented in PyTorch and run on 8 NVIDIA A800 GPUs (80GB each).

## C  EXPLORATORY COMPARISON WITH LARGE-SCALE GENERAL-PURPOSE RESTORATION MODELS

With the rapid development of diffusion models, large-scale variants trained on massive datasets have demonstrated impressive capabilities across diverse image restoration tasks Esser et al. (2024); Yu et al. (2024); Ai et al. (2024). These models, equipped with hundreds of millions or even billions of parameters, form the backbone of several general-purpose restoration frameworks that achieve state-of-the-art results in super-resolution, denoising, and image quality enhancement. Motivated by

their success, we further investigate whether such models can leverage their scale and training data to generalize to real-world deblurring.

However, our experiments reveal notable limitations. Using SUPIR Yu et al. (2024)as a representative model, we find that while it excels in enhancing perceptual quality—sometimes even surpassing ground-truth images in conventional quality metrics—it fails to effectively handle blur. As shown in 7, SUPIR struggles even on GoPro, one of the simplest synthetic benchmarks for motion deblurring, producing visually sharp but still blurred outputs. More strikingly, Fig. X also illustrates its shortcomings on complex real-world blur, where artifacts and residual degradation remain prominent.

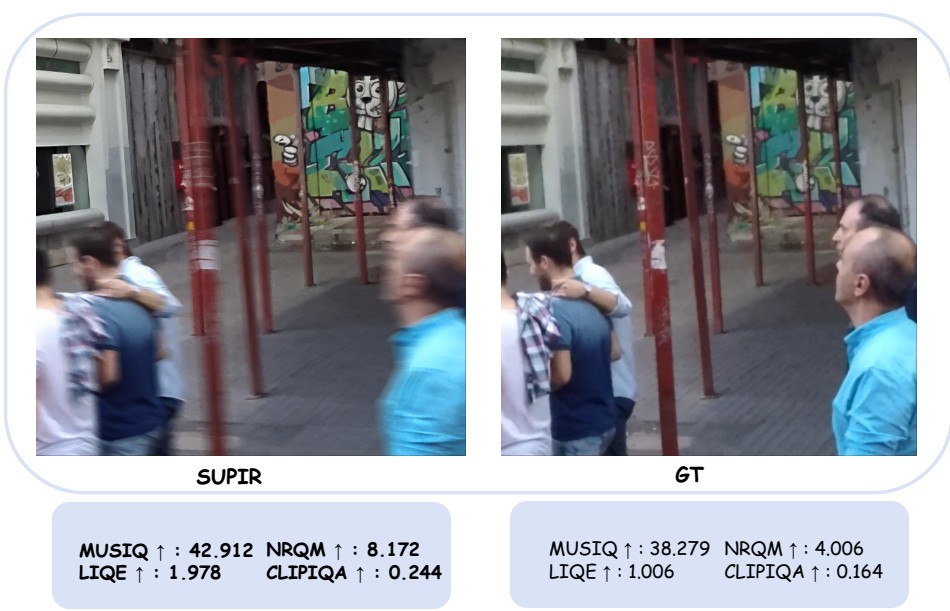

Figure 7: Qualitative and quantitative comparison of SUPIR and GT on GoPro, showing quality scores exceeding GT but failure to remove blur.

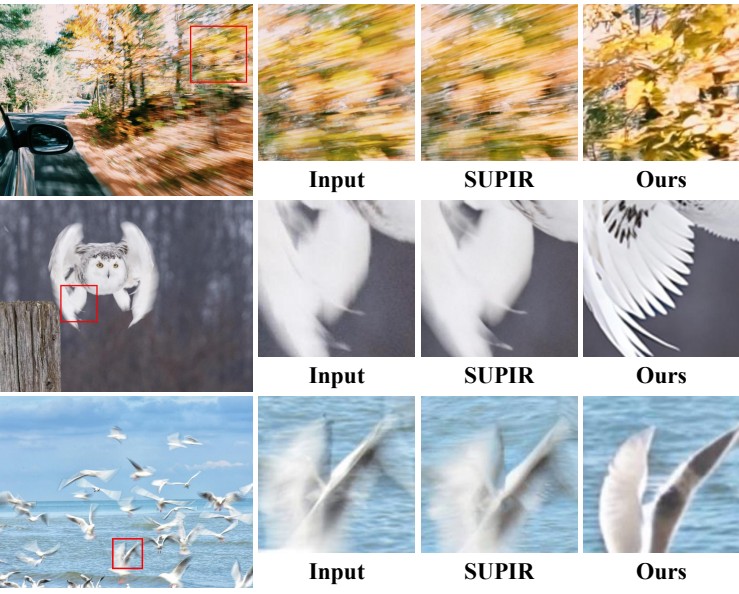

Figure 8: Quantitative comparison with SUPIR across real-world datasets.

These observations highlight an important gap: despite their remarkable success in other restoration tasks, large-scale diffusion models are not inherently equipped to handle the structural complexity of blur. This contrast further validates the necessity of explicitly modeling blur priors, as pursued in our proposed framework, to achieve robust and generalizable deblurring in real-world scenarios.

## D    DECLARATION OF USE OF LARGE LANGUAGE MODELS (LLM)

We confirm that this paper was written primarily by the authors. Large Language Models (LLMs) were used only as general-purpose tools for language refinement, including grammar correction and stylistic polishing. In particular, GPT-5 (OpenAI, 2025) was employed for minor rephrasing to improve clarity and readability. No LLM was involved in research ideation, experimental design, data analysis, or generation of substantive content.

## E    ADDITIONAL VISUAL RESULTS

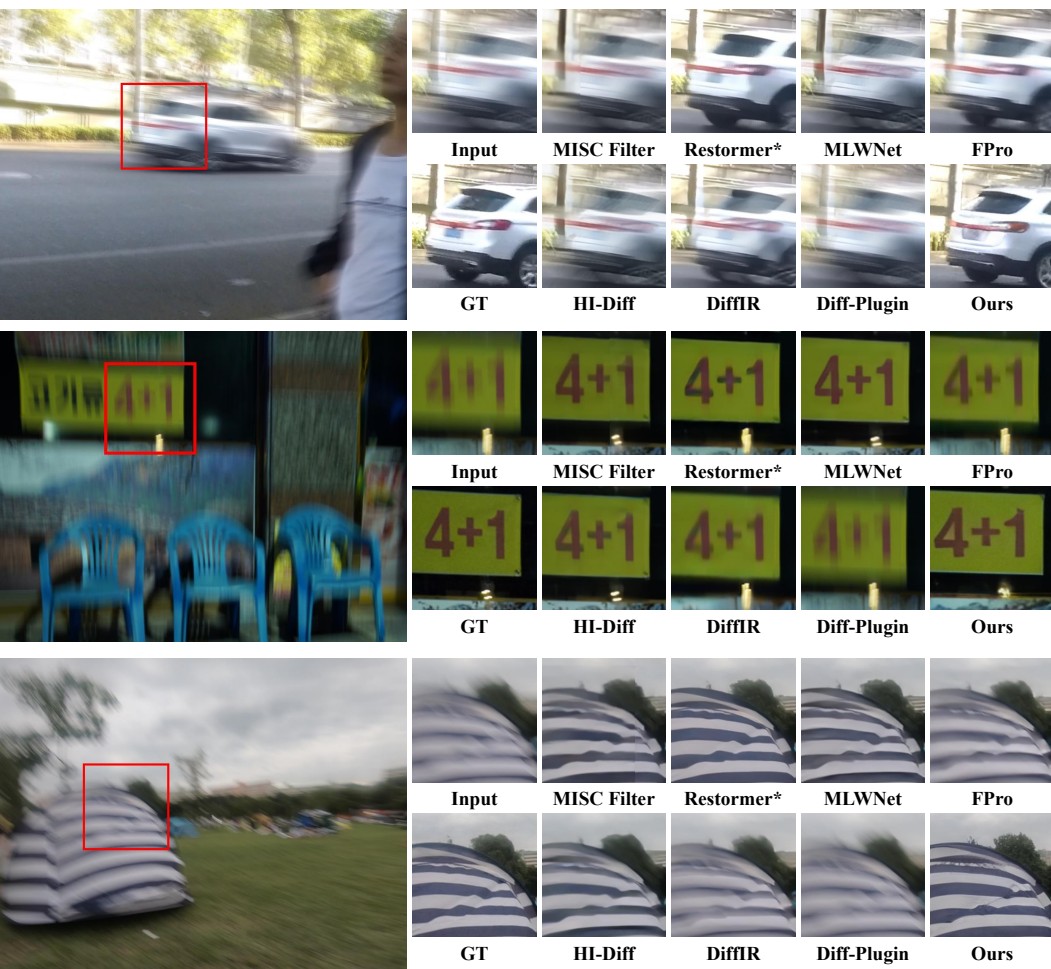

Figure 9: Qualitative comparison on HIDE, Realblur, and REDS (From top to bottom). GLOWDeblur effectively handles diverse blur patterns with high-quality restorations.

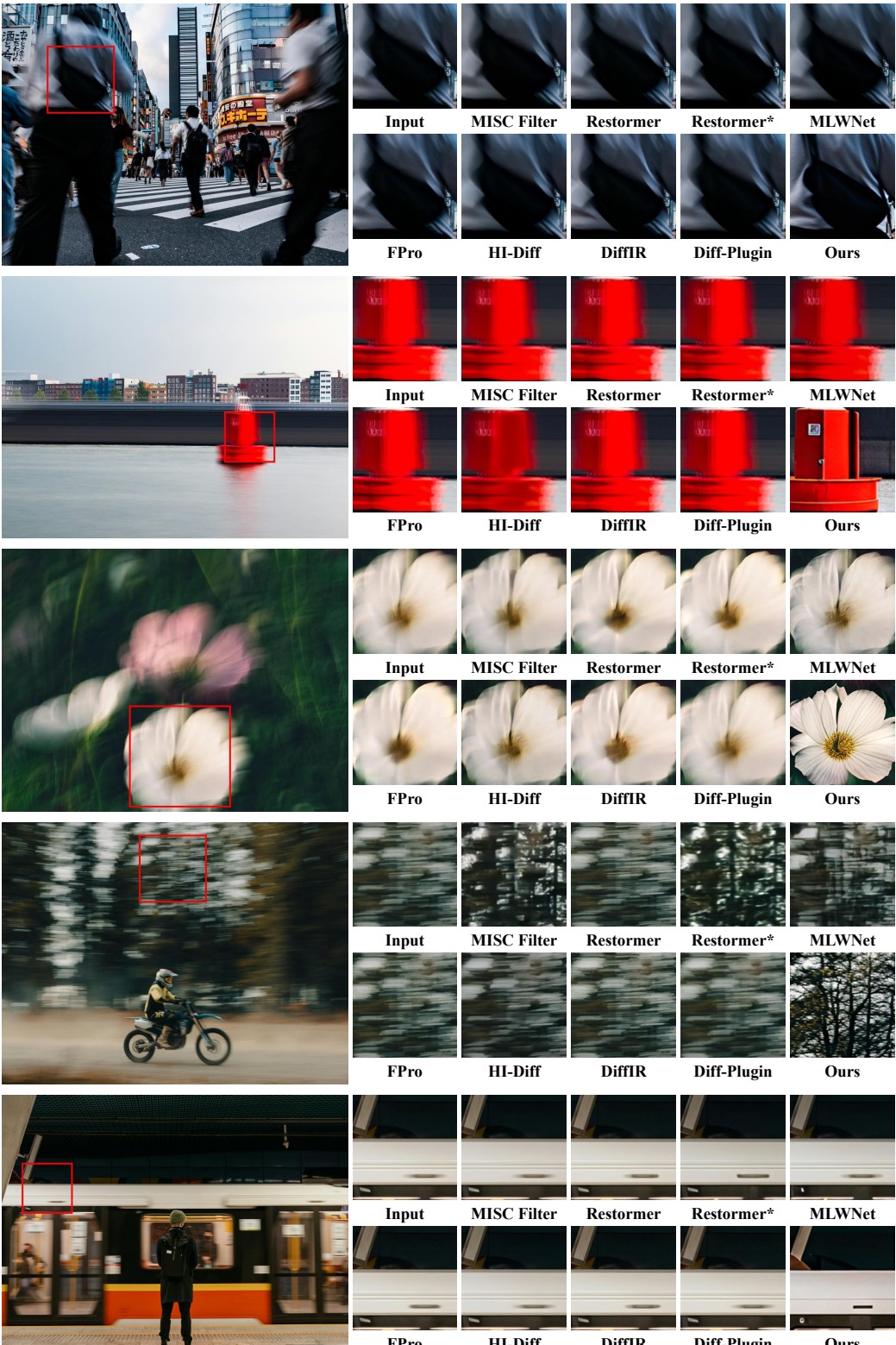

Figure 10: Comparison with SOTA deblur methods across real-world datasets.

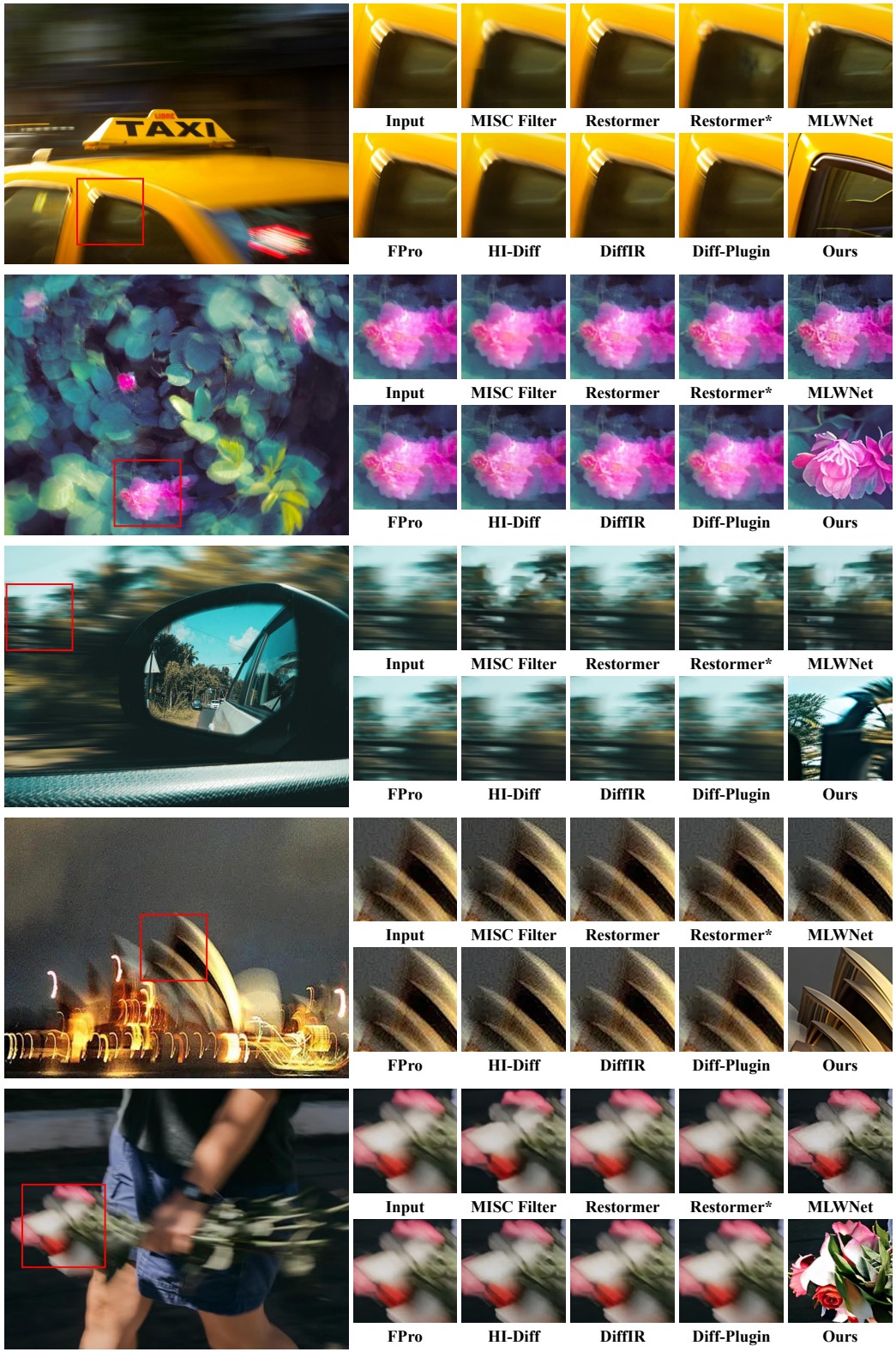

Figure 11: Comparison with SOTA deblur methods across real-world datasets.

