# OpenReview forum: "Toward Generalizable Deblurring: Leveraging Massive Blur Priors with Linear Attention for Real-World Scenarios"
_ICLR.cc/2026/Conference — ICLR 2026 Conference Withdrawn Submission_

### Official Review · Reviewer_npVR · 2025-10-26

**Soundness:** 3
**Presentation:** 3
**Contribution:** 2
**Rating:** 4
**Confidence:** 5

**Summary:**

This paper proposes a generalizable approach to real-world image deblurring. Observing that existing models suffer significant performance degradation on unseen data, the authors identify the core issue as the lack of diversity in blur patterns within training datasets, as well as an overreliance on pixel-wise losses that ignore structural and semantic consistency. To address this, they introduce a Blur Pattern Pretraining (BBP) strategy, which first learns blur priors from simulated datasets and then performs joint fine-tuning on real-captured data. They also present GLOWDeblur, a lightweight diffusion-based deblurring model that incorporates motion and semantic guidance to enhance restoration under severe blur. Experimental results across multiple benchmark and real-world datasets demonstrate the method’s superior generalization capabilities, highlighting the importance of blur priors and architectural design for practical deployment.

**Strengths:**

1.  Precise Problem Definition with Rigorous Evidence: The paper's motivation is clear and highly persuasive. It moves beyond the conventional discussion of "realism" by skillfully combining Table 1 (performance degradation) and Figure 3 (imbalanced pattern distributions). This robustly proves that the root cause of generalization failure is the biased and imbalanced distribution of "blur patterns" in the training data.

2.  Shifting Research Focus with a Systematic Strategy: Based on this key insight, the paper successfully shifts the research focus from pursuing "realism" to pursuing "blur pattern diversity." It proposes a systematic data strategy to address this issue. It first learns blur priors from simulation datasets with broad pattern coverage and then jointly fine-tunes on real-world datasets to align the distribution.

3. Superior Visual Results: This strategy directly targets the core of the problem, and its effectiveness is intuitively validated by the superior visual results.

**Weaknesses:**

1. Weak Methodological Innovation and Unaddressed Concerns: The paper's novelty is limited, primarily relying on stacking existing modules like MoG and SeG. These additions also introduce significant concerns: the accuracy of MoG's motion estimation is unverified and risks misguiding the restoration. Furthermore, the usage of SeG is ambiguous; if a VLM is required at test time, it introduces an unfair external annotation, and its robustness on low-quality, poorly-described images is unexplored.

2. Contradictory Metrics and Fidelity Concerns: The experimental results show a clear contradiction. The method excels in No-Reference (NR-IQA) metrics like MANIQA, yet it significantly lags behind in traditional Full-Reference (FR-IQA) metrics such as PSNR and SSIM. This strongly suggests that the model sacrifices fidelity for perceptual quality, likely suffering from the common "hallucination" problem in diffusion models (generating plausible but inaccurate details). The critical omission of the LPIPS metric fails to resolve these fidelity concerns.

3. Misleading "Lightweight" Claim: The paper's claim of being "lightweight" is misleading. It selectively highlights optimizations in the diffusion core while ignoring the total system complexity (including the UNet, MoG, and potentially a large VLM for SeG). The lack of FLOPs or actual inference time figures, combined with the mention of high-end training hardware (8x A800 80G GPUs), makes the claims of efficiency highly questionable.

**Questions:**

1.  Is the SeG module required to run during the inference stage? If so, does this imply the model relies on a large external VLM to achieve its advantage, leading to an unfair comparison and contradicting the "lightweight" claim?

2.  Is the motion estimation within the MoG module trained end-to-end, or does it use a fixed pre-trained model? If the former, are there specific loss functions or visualizations to prove it generates correct motion guidance? If the latter, how is its accuracy and applicability to your dataset ensured?

3.  How do the authors explain the significant gap where the model lags in PSNR/SSIM but excels in NR-IQA metrics? Does this imply the model suffers from the common "hallucination" issue (deviating from ground truth)? Why is the critical LPIPS metric missing to validate the perceptual fidelity against the ground truth?

4.  To validate the "lightweight" claim, what are the average inference time, the number of the parameters, and the FLOPs for the proposed method?

5.  Could you please detail the collection and filtering criteria for the self-created RWBlur400 dataset? Does this dataset have corresponding ground truth? If not, is relying solely on NR-IQA metrics sufficient to claim robust real-world performance?

---

### Official Review · Reviewer_joud · 2025-10-28

**Soundness:** 3
**Presentation:** 3
**Contribution:** 3
**Rating:** 4
**Confidence:** 4

**Summary:**

1. This paper discussed the importance of training deblurring methods with diverse blur patterns, and proposed a training strategy that first pre-trains the model on diverse synthetic blur pattern then real blurred imags.
2. A novel deblurring architecture consists of lighweight diffusion model is also proposed.

**Strengths:**

1. Performance shows that the proposed training strategy is helpful.
2. The lightweight diffusion architecture is novel, which is a good tradeoff between generalization and efficiency,

**Weaknesses:**

1. I think the paper should specifies that it is proposed to handling motion blur instead of general blurs.
2. The training pipeline is complex and hard to reproduce.
3. No explicit inference time evaluation.

**Questions:**

1. In line 144 and 129, I guess BBP is not the abbreviation of Blur pattern pretraining
2. How to get the blur pattern statistics shown in Table 3.
3. Were captions generated on degraded images or sharp images? If it is the latter, how to get correct captions during inference?
4. Is such an aggressive encoder able to reconstruct original images?
5. I know authors claimed that mix-training is sub-optimal, but I wonder the performance of training the model with just one step using all datasets mentioned in this paper.
6. In the owl of Figure 6, I wonder if the feather on the wing is really the reconstructed details or just hallucination. If it is hallucination, then it is dangerous to use it in camera.
7. How were other compared methods trained, on the same dataset or simply gran the pre-trained ckpt?
8. If the architecture is one of the contribution, then even without the BPP it should performs better than other methods. I would like to see more results about this.
9. Any evaluations of inference time?

---

### Official Review · Reviewer_b7dR · 2025-10-30

**Soundness:** 3
**Presentation:** 3
**Contribution:** 2
**Rating:** 4
**Confidence:** 3

**Summary:**

This paper addresses the limited generalization of image deblurring models in real-world scenario. It identifies main challenges from blur diversity-realism tradeoff and proposes BPP to learn diverse blur priors from simulated data, transferring them to real domains via joint fine-tuning. To further improve robustness, the authors introduce Motion and Semantic Guidance (MoSeG), which leverages motion cues and semantic context to enhance restoration quality.

**Strengths:**

1. Writing: The paper is well-organized and clearly written, making it easy to follow.

2. Logical Idea: The integration of motion-related and semantic-related information for deblurring is intuitive and well-motivated, representing a logical extension of existing approaches. The overall framework design is cohesive, and experiments demonstrate promising results.

**Weaknesses:**

1. Motion Guidance: The proposed motion modeling appears limited to 2D directional blur, whereas real-world blur often includes depth-axis motion components. Consequently, BPP may fail to capture full 3D motion complexity, reducing its applicability to realistic scenarios. Moreover, since motion trajectories require paired sharp images to be computed, the motion guidance component can only be trained on synthetic datasets, potentially restricting its generalization to real-world data.

2. Semantic Guidance: Under conditions of severe blur, the pretrained VLM~(QwenVL) model used for semantic extraction may produce inaccurate or unreliable outputs, weakening its contribution to deblurring quality.

3. Typo: BBP -> BPP

**Questions:**

1. Could the authors provide a comparison with models trained on all major datasets (GoPro, HIDE, REDS, RealBlur, etc.) to more convincingly demonstrate the effectiveness of their approach?

2. What is the patch size or spatial scale used for motion estimation in motion guidance, and how sensitive is the performance to this choice?

---

> ### Author Response · Authors · 2025-11-14
> **Rebuttal to Reviewer b7dR**
>
> We thank the reviewer for the constructive comments and for recognizing the clarity and logical motivation of our framework.
>
> 1. On the concern about 2D motion guidance and depth-axis components.
>
> We agree that depth-axis motion can introduce additional cues. However, in the context of motion blur formation, its effect is primarily reflected in the \textbf{effective blur magnitude} observed on the 2D projection plane. Under the standard pinhole camera model, a 3D point $(X, Y, Z)$ with instantaneous motion $(\Delta X, \Delta Y, \Delta Z)$ during exposure is projected onto image coordinates as
> $$
> x = f \frac{X}{Z}, \quad y = f \frac{Y}{Z}.
> $$
> The corresponding projected motion trajectory is
> $$
> \Delta x = f\left(\frac{\Delta X}{Z} - \frac{X \,\Delta Z}{Z^{2}}\right), \quad
> \Delta y = f\left(\frac{\Delta Y}{Z} - \frac{Y \,\Delta Z}{Z^{2}}\right).
> $$
>
> It can be seen that the \textbf{depth-axis motion} $\Delta Z$ primarily affects the blur \textbf{magnitude} via the depth-dependent scaling factor $1/Z$. That is, for identical 3D motion amplitude $s$, objects farther from the camera exhibit proportionally smaller projected motion magnitudes $|\Delta x|$ and $|\Delta y|$. This means that, while full 3D modeling does contain extra information, the \emph{dominant observable factor} in the 2D blurred image is still the \textbf{trajectory direction and magnitude}---the exact cues our 2D motion guidance module is designed to capture.
>
> Importantly, while we acknowledge the additional information embedded in full 3D motion, its effect on image formation is \textbf{largely manifested through the depth-dependent scaling of the projected blur magnitude}, as shown in the equations above. In other words, the depth-axis component is already implicitly encoded in the observable 2D blur amplitude. This makes 2D motion trajectories a sufficiently expressive representation for the deblurring objective. Introducing explicit 3D motion estimation would substantially increase task difficulty and model complexity, yet provide limited additional benefit for this specific problem, and could even \textbf{negatively impact generalization and computational efficiency}, which are central goals of our framework. Although our motion trajectories are trained on synthetic data, the learned blur-pattern priors remain dataset-agnostic and transfer well to real-world scenarios, as consistently validated in our cross-domain experiments and ablations.
>
> 2. On the concern about semantic guidance reliability under severe blur.
>
> We acknowledge that vision--language models can become less precise under extreme degradation. However, even in challenging cases, pretrained VLMs (e.g., Qwen-VL) retain strong robustness and can still extract \textbf{global semantic context} (e.g., object category, coarse layout), which is exactly what we need. The semantic guidance is \emph{not} intended to provide pixel-level accuracy; rather, it offers stable high-level priors that help the diffusion model restore severely degraded regions. This is fundamentally different from classical deblurring methods, which must rely solely on the corrupted input and therefore struggle with real-world generalization. Our experiments show that even coarse semantic cues significantly benefit structure recovery.
>
> We appreciate the reviewer’s suggestions and will integrate these clarifications into the final version.

---

### Official Review · Reviewer_CQv5 · 2025-10-30

**Soundness:** 2
**Presentation:** 3
**Contribution:** 2
**Rating:** 2
**Confidence:** 4

**Summary:**

This paper tackles the generalization problem in image deblurring caused by the biases in deblurring datasets. The authors identified the characteristics in the current datasets and investigated the generalization performance using the existing deblurring method, Restormer. Then, this paper proposes Blur Pattern Pretraining to use a simulation-based dataset, GSBlur, to pretrain the deblurring models, thus improving the generalization ability. Furthermore, the deblurring performance is enhanced by the proposed GLOWDeblur model, which consists of several auxiliary tasks like motion estimation, text-guided diffusion.

**Strengths:**

- The paper is well-motivated. The generalization problem in the image deblurring task is crucial and has not been fully explored.
- The characteristic analysis of the current deblurring datasets contributes to the deblurring community.
- The paper is well-written and easy to follow.

**Weaknesses:**

- There are concerns for the BBP in tackling the generalization problem: BBP relies on GSBlur, a larger existing simulation dataset. Thus, the improvement may be due to 1) GSBlur is a larger training dataset and 2) GSBlur covers the blur characteristics of each test set, rather than generalizing to new blur characteristics.
- GLOWDeblur significantly degrades the PSNR and SSIM in most cases in Table 3. However, these are major metrics in the image deblurring task. This method may severely affect the pixel-level similarity between the output and the ground truth, which is not a satisfactory deblurring result.
- Many auxiliary tasks (motion estimation, text-guided diffusion generation) are equipped into the model. This raises the concern of the efficiency compared with other methods. But this is not mentioned in the paper.

**Questions:**

Please refer to the weakness part.

---

> ### Author Response · Authors · 2025-11-14
> **Rebuttal to Reviewer CQv5**
>
> We thank the reviewer for the thoughtful comments. Regarding the concerns on BPP and GSBlur, our paper already clarifies the data organization of BPP: GSBlur is only *one* component within the pretraining stage. BPP additionally includes simple yet diverse synthetic blur augmentations to further enrich the blur patterns. The purpose is not to rely on GSBlur “covering” the test distributions, but to leverage its *pattern diversity* to learn transferable blur-pattern priors. Our cross-dataset results in Table 1 further support this point: if GSBlur truly covered each test set, training on GSBlur alone would already yield strong cross-dataset performance, yet clear domain gaps remain. This indicates that GSBlur is still far from real captured images in terms of visual quality, noise, and texture statistics, but its diverse blur patterns—despite being lower quality—show strong potential for bridging dataset gaps and alleviating the mismatch between existing datasets and real-world blur. This is precisely the motivation behind proposing BPP: we use GSBlur not because it matches real-world distributions, but because its rich and varied blur patterns provide a generic prior when combined with limited real data. We will revise the text to emphasize this rationale more explicitly.
>
> Concerning the drop in PSNR/SSIM, we agree these metrics are important, yet recent image restoration research increasingly recognizes that distortion-based metrics are often misaligned with human perception, especially under severe real-world blur where exact pixel-wise recovery is inherently ill-posed. Our method targets improving perceptual and semantic fidelity, which is also reflected in the metrics and qualitative comparisons provided. GLOWDeblur produces results with sharper structures and more realistic textures, addressing the primary shortcomings of PSNR-focused models that tend to oversmooth and lose details. We will clarify that the goal of our model is to enhance perceptual quality and robustness under distribution shift, a direction now widely adopted across restoration tasks.
>
> On the efficiency concern, our architectural choices are made to mitigate the inherent computational cost of diffusion-based models for practical deployment. Our notion of being “lightweight” refers specifically to these design decisions that reduce diffusion overhead, and is not a claim that our method is faster than all comparison baselines, as clearly discussed in the paper. Diffusion models are naturally more expensive, and our intention is to reduce this cost—not to contradict this property. To this end, we employ linear attention instead of quadratic attention and adopt a 32× deep-compression autoencoder rather than a standard 8× VAE, substantially lowering the runtime burden. Moreover, the auxiliary tasks (motion and semantic guidance) are injected into the DiT through simple concatenation, instead of introducing complex ControlNet-style conditional branches, ensuring minimal additional overhead. We will emphasize this point more clearly in the revised version and additionally provide comparisons of parameter count and inference speed with and without these lightweight optimization strategies.

---

### Note · Authors · 2025-11-14

**Comment:**

We respectfully request the withdrawal of this submission. Following extensive internal deliberation, the authors have decided to further develop and substantially extend the scope of the work, with the intention of submitting a revised manuscript to a more appropriate venue. All co-authors unanimously support this withdrawal request. We thank the program committee for their time, consideration, and efforts in handling our submission.

**Withdrawal Confirmation:**

I have read and agree with the venue's withdrawal policy on behalf of myself and my co-authors.